# Alisma Orientalis Extract Ameliorates Hepatic Iron Deregulation in MAFLD Mice via FXR-Mediated Gene Repression

**DOI:** 10.3390/nu16142272

**Published:** 2024-07-15

**Authors:** Yanlin Li, Ke Zhang, Yue Feng, Lei Wu, Yimin Jia, Ruqian Zhao

**Affiliations:** 1MOE Joint International Research Laboratory of Animal Health & Food Safety, Nanjing Agricultural University, Nanjing 210095, China; 13592662352@163.com (Y.L.); 19825014155@139.com (K.Z.); yuefeng@cau.edu.cn (Y.F.); leiwu@njau.edu.cn (L.W.); jymrobin@hotmail.com (Y.J.); 2Key Laboratory of Animal Physiology & Biochemistry, College of Veterinary Medicine, Nanjing Agricultural University, Nanjing 210095, China; 3National Key Laboratory of Meat Quality Control and Cultured Meat Development, Nanjing 210095, China

**Keywords:** alisma extract, MAFLD, lipid metabolism, iron metabolism, FXR

## Abstract

Iron is a vital trace element for our bodies and its imbalance can lead to various diseases. The progression of metabolic-associated fatty liver disease (MAFLD) is often accompanied by disturbances in iron metabolism. Alisma orientale extract (AOE) has been reported to alleviate MAFLD. However, research on its specific lipid metabolism targets and its potential impact on iron metabolism during the progression of MAFLD remains limited. To establish a model of MAFLD, mice were fed either a standard diet (CON) or a high-fat diet (HFD) for 9 weeks. The mice nourished on the HFD were then randomly assigned to the HF group and the HFA group, with the HFA group receiving AOE by gavage on a daily basis for 13 weeks. Supplementation with AOE remarkably reduced overabundant lipid accumulation in the liver and restored the iron content of the liver. AOE partially but significantly reversed dysregulated lipid metabolizing genes (SCD1, PPAR γ, and CD36) and iron metabolism genes (TFR1, FPN, and HAMP) induced by HFD. Chromatin immunoprecipitation assays indicated that the reduced enrichment of FXR on the promoters of SCD1 and FPN genes induced by HFD was significantly reversed by AOE. These findings suggest that AOE may alleviate HFD-induced disturbances in liver lipid and iron metabolism through FXR-mediated gene repression.

## 1. Introduction

Iron is primarily stored in human red blood cells, liver, bone marrow, and spleen, playing indispensable roles in the synthesis of hemoglobin [1], oxygen transport [1], body immunity [2], and neural signal transmission [3]. The liver serves as an indicator of total body iron [4], while hepatocytes secrete the protein hepcidin into the bloodstream, regulating systemic iron uptake and thus governing systemic iron homeostasis [5,6]. Several liver diseases can lead to an imbalance in liver iron metabolism, such as Toxoplasma gondii infection [7], bacterial infection [8], and thalassaemia [9].

Metabolic syndrome, encompassing obesity and metabolic-associated fatty liver disease (MAFLD), is primarily characterized by dysregulated lipid metabolism, often accompanied by hepatic iron homeostasis imbalance [10]. The changes in hepatic iron levels exhibit distinct patterns at different stages. A high-fat diet increases liver iron content for less than 8 weeks [11,12]. Between 10 to 12 weeks, the iron content of the liver is not remarkably affected by a high-fat diet [13,14,15] or may result in an increase [16]. However, a high-fat diet lasting more than 16 weeks leads to a decrease in liver iron content [15,17,18]. Lipid accumulation occurs when the pathways for lipid uptake exceed the pathways for lipid disposal [19], which involves lipid de novo synthesis, lipid uptake, β-oxidation, and the efflux of triglyceride-laden very-low-density lipoprotein (VLDL). Iron homeostasis is primarily determined by iron intake, transport, storage, and excretion. The transferrin-bound iron (TBI) and non-transferrin-bound iron (NTBI) in the blood circulation enter cells via transferrin receptors (TFRs) [20,21,22] and ZIP14 [23], respectively. Inside the cells, iron is transported out of the endosome via divalent metal transporter 1 (DMT1), stored in ferritin, and released back into the bloodstream via ferroportin (FPN) [23].

Farnesoid X receptor (FXR), a widely expressed nuclear receptor, plays a central role in regulating bile acid, lipid, and glucose homeostasis [24]. It serves as a crucial regulatory target for alleviating MAFLD [25], with many clinical trials focusing on modulating MAFLD by activating FXR [26]. Moreover, the relationship between FXR and hepatic iron metabolism is intriguing. Systemic FXR knockout mice exhibited exacerbated cisplatin-induced hepatic iron elevation [27]. Hepatic iron was significantly increased in FXR-deficient mice compared to controls in a model of iron overload induced by a high-iron diet [28]. Notably, FXR protein content is reduced in high-fat diet animal models [28,29]. However, the precise involvement of FXR in liver lipid metabolism and iron metabolism remains unclear.

Alisma orientale (Sam.) is a traditional Chinese medicine known for its anti-uric, anti-diuretic, anti-inflammatory, and anti-atherosclerotic effects [29,30,31]. Triterpenes derived from Alisma orientale act as agonists of the FXR receptor [32]. Alisma orientale’s potential to alleviate MAFLD has been demonstrated in several studies [33,34,35]. Alisma orientale alcohol extract and its monomer alisol B 23-acetate have been shown to prevent hepatic steatosis induced by a Western diet [33] and methionine-choline-deficient (MCD) diet [36] by activating FXR. Additionally, alisol B alleviated lipid steatosis and inflammation induced by a high-fat and carbon tetrachloride mixture in mice through the RXRα-PPARγ-CD36 pathway [37]. However, whether and how Alisma orientale relieves hepatic lipid deposition and iron metabolism disorder induced by high-fat diet has not been reported.

Here, we established the MAFLD mouse model to assess the impact of Alisma orientalis extract (AOE) on lipid metabolism and iron metabolism induced by a high fat diet. Additionally, we investigated the regulatory role of FXR in maintaining lipid and iron metabolism homeostasis. Collectively, our study identifies new targets for MAFLD treatment with AOE, paving the way for innovative drug development avenues.

## 2. Materials and Methods

### 2.1. Plant Extract

Alisma orientale was purchased from Nanjing Hospital of Integrated Traditional Chinese and Western Medicine (Nanjing, China) and ground into powder. The ratio of Alisma to ethanol was 1:10 (*w*:*v*). The power was refluxed with 85% ethanol three times, each time for 1 h. The extracts were then combined and concentrated using a rotary evaporator. The contents of triterpenoid alcoholic ketones were determined using the Zimmerman method and the extract was administered by gavage at a concentration of 72 mg/kg [35]. Subsequent laboratory analysis showed that the extract consisted of alisol A, alisol B, alisol B 23-acetate, and alisol A 24-acetate with concentrations determined by HPLC to be 0.9 mg/g, 0.7 mg/g, 0.07 mg/g, and 0.87 mg/g, respectively.

### 2.2. Animals and Treatment

Approved by the Animal Ethics Committee of Nanjing Agricultural University (protocol code PZ2018019, 11 December 2018), all procedures related to animals were conducted in accordance with the ‘Guidelines on Ethical Treatment of Experimental Animals’ (2006, No. 398) established by the Ministry of Science and Technology of China. Four-week-old SPF C57BL/6 mice were procured from the Center for Comparative Medicine at Yangzhou University and housed at the Laboratory Animal Centre of Nanjing Agricultural University under controlled conditions (22 ± 0.5 °C, 50 ± 5% humidity, and a 12-h light-dark cycle). Following a one-week acclimatization period, the mice were randomly assigned into two groups. One group received a control diet providing 10% of energy from fat (CON, *n* = 8), while the other group received a high-fat diet (HFD) providing 60% of energy from fat (*n* = 16) for a duration of 9 weeks. Impaired glucose tolerance and weight gain of more than 20% indicated successful establishment of MAFLD. Appendix A shows the nutritional composition of the experimental diets. The HFD group was further divided, with one subgroup receiving Alisma orientale extract (AOE) (content of triterpenoid alcoholic ketones 72 mg/kg/day), while the other subgroup continued on the high-fat diet and received the same amount of water by gavage. Experimental treatments were carried out for 13 weeks, during which the control group was fed a normal diet and received water by gavage, similar to the HFD group. At the end of the 13th week, the mice were subjected to an overnight fast before being euthanized. Blood was collected and plasma was isolated and kept at −20 °C for later analysis. Liver samples were collected and stored at −80 °C.

### 2.3. Plasma Biochemical Parameter Determination

The levels of the following biochemical indicators in plasma were measured using an automatic biochemical analyzer (Hitachi 7020, Tokyo, Japan). The reagents used were purchased from Meikang Bioengineering Co., Ltd. (Shanghai, China).

Here are the specific biochemical reagent details: plasma alanine aminotransferase (ALT) activity (H001), plasma iron level (H309W), triglycerides (TG) (H201), high-density lipoprotein cholesterol (HDL-c) (H203), nonesterified fatty acids (NEFA) (OD444), low-density lipoprotein cholesterol (LDL-c) (H207), total cholesterol (Tch) (H202), and total iron-binding capacity (TIBC) (H465).

### 2.4. Histological Evaluation of Liver Tissues

Five liver tissue samples from each group were embedded, sectioned, and stained with H&E and Oil Red using classical histological methods in the laboratory [17]. The steps for Prussian blue staining followed the protocol reported in the literature [18].

### 2.5. Determination of Hepatic Triglyceride Content and Hepatic Total Cholesterol

A 30 mg liver tissue sample from each mouse was placed in an EP tube and homogenized using tissue lysis buffer, following the manufacturer’s protocol for quantification. Hepatic triglyceride (TG) levels (A110-1-1) and total cholesterol (Tch) levels (A111-1-1) were measured using commercial assay kits purchased from Nanjing Jiancheng Bioengineering Institute (Nanjing, China).

### 2.6. Quantification of Hepatic Iron Content

The hepatic iron content was assessed utilizing graphite atomic absorption spectrophotometry and presented as μg/g wet tissue. Our methodology for determining liver iron content closely followed the procedure outlined in our prior publication [18].

### 2.7. Determination of Hepatic MDA Content and SOD Activity

Commercial assay kits for superoxide dismutase (SOD) (A001-3) and malondialdehyde (MDA) (A003-1) were purchased from Nanjing Jiancheng Bioengineering Institute. The assays were performed according to the instructions provided with each kit.

### 2.8. Total RNA Isolation and Real-Time PCR

A 30 mg liver tissue sample from each mouse was placed in a homogenization tube with three sterile beads. Total RNA was extracted using TRIzol reagent (Invitrogen, Carlsbad, CA, USA) according to the manufacturer’s instructions. After verifying the quality of the RNA, reverse transcription was performed using a reverse transcription kit (R223-01, Vazyme, Nanjing, China). The target gene mRNA sequences were obtained from the GenBank database on the NCBI website, and primers (Appendix A) were designed accordingly. Quantitative real-time PCR was performed using the QuantStudio 6 Flex Real-Time PCR System (Applied Biosystems, Foster City, CA, USA). The data were analyzed using the 2^−ΔΔCt^ method.

### 2.9. Western Blot Analysis

Thirty milligrams of mouse liver tissue were homogenized in protein lysis buffer using a precooled automated homogenizer. Protein concentration was determined using the Pierce BCA Protein Assay Kit (Thermo Fisher, Waltham, MA, USA) following previously reported protocols for tissue extraction and protein quantification [18]. Antibodies were used for Western blot analysis (Appendix A).

### 2.10. ChIP Assay

Chromatin immunoprecipitation (ChIP) was conducted following established procedures [38]. In summary, frozen liver samples were pulverized in liquid nitrogen and then mixed with PBS containing protease inhibitors. Subsequently, 37% formaldehyde was added to cross-link the intracellular DNA and proteins and the reaction was terminated using 2.5 M glycine. The resulting pellets were lysed and sonicated to an average fragment size of approximately 300 bp. The protein–DNA complexes were diluted with ChIP lysis buffer and incubated overnight at 4 °C with FXR antibody. A separate set was used as a negative control with normal IgG or without any antibody. The next day, Protein G agarose beads (sc-2003, Santa Cruz Biotechnology, Dallas, TX, USA) were used to capture the immunoprecipitated chromatin complexes. Reverse cross-linking was carried out at 65 °C for 5 h to release DNA fragments from the immunoprecipitated complexes, followed by DNA purification. Putative Farnesoid X Receptor-Responsive Elements (FXREs) in the promoters of SCD1 and FPN were predicted using JASPAR 2024 (http://jaspar.genereg.net, accessed on 1 February 2024). Primers were designed to amplify the sequences containing these putative FXREs, as listed in Appendix A.

### 2.11. Statistical Analysis

This study used nonparametric tests. The Kruskal–Wallis test was used to determine the significance of differences between the three groups. If the Kruskal–Wallis test confirmed significant differences, the Kolmogorov–Smirnov test in SPSS 20.0 software (SPSS Inc., Chicago, IL, USA) was used to assess the normal distribution of the data or log-transformed data. Data that met the criteria for normal distribution were analyzed using one-way analysis of variance (ANOVA) with the Least Significant Difference (LSD) test, while nonparametric data were analyzed using the Kruskal–Wallis or Mann–Whitney U test. Data are presented as 41 medians ± interquartile ranges. Statistical significance is defined as * *p* < 0.05 and ** *p* < 0.01. All bar plots in our study were generated by GraphPad Prism (version 9.0.0, GraphPad Software, Boston, MA, USA).

## 3. Results

### 3.1. AOE Alleviated HFD-Induced Lipid Deposition in Liver

To evaluate whether AOE can alleviate high-fat diet-induced MAFLD in mice, we measured the apparent indicators, plasma biochemical indicators, and liver-related markers in the mice. Compared with the CON group, body weight and liver weight as well as the liver-to-body weight ratio increased significantly in the HF group (*p* < 0.01). The administration of AOE significantly (*p* < 0.01) inhibited HFD-induced growth in body weight, liver weight, and liver ratios (Figure 1A–C). Plasma biochemical tests showed that, compared with the CON group, the HF group had significantly (*p* < 0.01) higher levels of nonesterified fatty acid (NEFA), total cholesterol (Tch), high-density cholesterol (HDL-C), and low-density cholesterol (LDL-C). Compared with the HF group, the HFA group significantly (*p* < 0.05) reduced the level of NEFA, Tch, and LDL-C in plasma (Figure 1D,E). Histopathological evaluation, liver injury assessment, and hepatic triglyceride content analysis revealed that, compared to the CON group, the HF group exhibited a significant (*p* < 0.01) increase in hepatic lipid droplet accumulation, liver injury, and elevated hepatic triglyceride levels. Oral administration of AOE significantly (*p* < 0.01) reduced hepatic lipid droplet accumulation, mitigated liver injury, and decreased hepatic triglyceride content. (Figure 1F–I). The supplementary administration of AOE also reversed (*p* < 0.05) the HFD-induced elevation of total cholesterol (Tch) (Figure 1J). These results indicate that AOE successfully alleviated HFD-induced lipid deposition in the liver.

### 3.2. AOE Alleviated HFD-Induced Iron Homeostasis Disruption in Liver

The process of MAFLD is accompanied by the occurrence of iron metabolism disorders [16,18]. HFD significantly (*p* < 0.05) increased the concentration of transferrin-bound iron and decreased (*p* < 0.05) the unsaturated iron-binding capacity (UIBC) in the plasma, without affecting the total iron-binding capacity (TIBC). Oral administration of AOE significantly reversed the HFD-induced disruptions in plasma iron metabolism (Figure 2A). At the same time, AOE supplementation completely (*p* < 0.05) attenuated the HFD-induced decrease in the iron content of the liver (Figure 2B,C). Iron imbalance leads to excessive oxygen ion production, resulting in oxidative stress. We further explored oxidative stress-related indicators in the liver. Supplementing AOE partially (*p* < 0.05) alleviated the increase in malondialdehyde (MDA) content and superoxide dismutase (SOD) activity induced by HFD as it also alleviated (*p* < 0.05) the increase in gene expression levels related to oxidative stress (Figure 2D–F). These results effectively support the phenomenon that AOE alleviates iron homeostasis imbalance (*p* < 0.05). Overall, these results indicate that AOE significantly alleviated liver iron metabolism disorders during the progression of MAFLD.

### 3.3. AOE Alleviated HFD-Induced Lipid-Metabolic Gene Expression

To further investigate the alleviating effects of AOE on lipid deposition, we examined the changes in lipid metabolism-related genes. At the mRNA level, compared to the CON group, the expression of the lipid uptake regulatory gene PPAR γ and its target gene CD36 as well as the lipogenic gene SCD1 were markedly (*p* < 0.01) upregulated in the liver of HFD-fed mice (Figure 3A). Oral administration of AOE markedly (*p* < 0.05) reduced the HFD-induced increase in hepatic mRNA expression of PPAR γ, CD36, and SCD1.

At the protein level, HFD significantly decreased the expression of PPAR γ, contrary to the mRNA level. Compared to the HF group, the HFA group showed a significant (*p* < 0.01) increase in PPAR γ protein levels (Figure 3B). The HFD markedly (*p* < 0.01) increased the protein expression levels of the lipid uptake gene CD36 (Figure 3C) and the lipogenic gene SCD1 (Figure 3D). Compared to the HF group, the HFA group showed a significant (*p* < 0.01) decrease in the protein levels of CD36 and SCD1. Oral administration of AOE did not reverse the high-fat diet-induced decrease in the protein levels of the lipid oxidation gene PPARα (Figure 3E). These results indicate that AOE alleviates HFD-induced lipid deposition in mice primarily by reducing the expression of lipid uptake and synthesis genes.

### 3.4. AOE Mitigated Disruption of Iron Metabolism Gene Expression Induced by HFD

To further investigate the alleviating effects of AOE on hepatic iron deficiency, we examined the changes in genes related to iron metabolism. FPN was markedly upregulated, while its regulatory gene HAMP was markedly decreased in the HF group (Figure 4A–C) at the level of mRNA. These dysregulations induced by HFD were markedly reversed by AOE supplementation (*p* < 0.05). At the protein level, transferrin receptor 1 (TFR1) was markedly decreased (Figure 4D), while FPN was markedly increased (Figure 4F) in the HF group. (*p* < 0.05). All changes induced by HFD were restored by AOE supplementation. There was no change in transferrin receptor 2 (TFR2) (Figure 4E). These results indicate that AOE primarily regulates iron homeostasis by modulating iron intake and excretion, thereby correcting iron imbalance.

### 3.5. AOE Rectified the HFD-Induced FXR Enrichment Decrease on the Promoters of Affected Genes

We have identified that AOE regulates specific target genes involved in lipid deposition and hepatic iron deficiency induced by HFD. However, the specific regulatory mechanisms remain unclear. Therefore, we investigated the protein level changes of several nuclear transcription factors that may be involved in MAFLD progression. A high-fat diet did not alter the protein content of RXR (Figure 5A), LXR (Figure 5B), or GR (Figure 5C) but significantly decreased the protein content of FXR (Figure 5D) compared with the CON group. AOE treatment partially restored FXR protein content in the liver (*p* < 0.05). ChIP-PCR analysis exposed alterations in FXR binding on the promoters of SCD1 and FPN genes. Following chromatin immunoprecipitation with an FXR antibody, purified DNA was used as a template to amplify fragments containing the putative FXREs on the promoters of SCD1 (Figure 5E) and FPN (Figure 5F). In the HF group, FXR binding to fragments 1 and 2 of the SCD1 gene promoter was noticeably decreased (*p* < 0.05), as was FXR binding to the FPN gene promoter fragment (*p* < 0.05). AOE treatment partially restored FXR enrichment on the SCD1 and FPN gene promoters. FXR not only regulates lipid metabolism in the progression of MAFLD [25,26] but also regulates iron metabolism [27,28]. Meanwhile, AOE and alisol B 23-acetate contained in AOE have been shown to prevent hepatic steatosis induced by a Western diet [33] and an MCD diet [36] by activating FXR. Therefore, we can conclude that AOE may reduce the expression of the lipid metabolism gene SCD1 and iron metabolism gene FPN by activating FXR, thereby alleviating lipid accumulation and iron metabolism disorders induced by HFD in mice.

## 4. Discussion

In this investigation, oral administration of Alisma orientalis extract (AOE) notably mitigated extravagant lipid accumulation and recovered disrupted iron balance in the livers of HFD-induced MAFLD mice. We identified several key genes associated with lipid and iron metabolism as targets of AOE intervention, including the de novo lipogenesis SCD1, lipid uptake genes CD36 and PPAR γ, and iron metabolic genes FPN and TFR1. Moreover, our findings indicate that the beneficial outcomes of AOE on dysregulation of hepatic lipid and iron homeostasis induced by a high-fat diet (HFD) are mediated through the FXR signaling pathway. This study enhances our insight into the mechanisms implicit in the therapeutic potential of AOE in MAFLD, shedding light on its promising role in modulating lipid and iron metabolism through FXR-related pathways.

Hepatic iron content shows variability based on the stage or seriousness of MAFLD development. In animal models triggered by an HFD for 8 weeks [11,12], there is a marked increase in iron in the liver deposition. In models fed an HFD for 10–12 weeks [13,14], no changes in hepatic iron content are observed. However, when HFD feeding exceeds 16 weeks [39,40], a significant reduction in liver iron levels is observed. The underlying reasons for these stage-dependent changes remain unclear and are likely to be related to the intricate correlations between various cell types within the liver. In this investigation, mice in the HF group were fed a high-fat diet (HFD) for 22 weeks. Consistent with previous studies, we observed a substantial decrease in hepatic iron content accompanied by a rise in lipid accumulation. Additionally, prior research [41] has shown an inverse relationship between hepatic iron content and plasma iron levels. The HF group mice presented a noteworthy increase in plasma transferrin-bound iron concentration.

The mechanisms leading to lipid deposition during MAFLD progression vary depending on different feeding durations, induction methods, and animal models. Some studies suggest that increased lipid uptake from the circulatory system and excessive de novo lipogenesis in hepatocytes are the primary causes of lipid deposition [42,43,44]. Other studies indicate that impaired fatty acid oxidation and disruption of the process of triglyceride export via very low-density lipoprotein (VLDL) are more critical factors contributing to lipid accumulation in hepatocytes [45,46,47]. Our study revealed that a high-fat diet markedly upregulated de novo lipid synthesis genes SCD1 and lipid uptake gene CD36, while simultaneously downregulating the fatty acid oxidation-related gene PPARα. Treatment with AOE attenuated the upregulation of SCD1 and CD36 without affecting PPARα expression. This aligns with previous findings where alisol B 23-acetate, derived from Alisma orientalis, mitigated lipid accumulation by suppressing SCD1 and FASN expression [36,48]. Moreover, Alisol B demonstrated therapeutic efficacy in murine MASH models by modulating the RXR-PPARγ-CD36 pathway [37]. Interestingly, our study observed a discrepancy between hepatic mRNA and protein expression of PPAR γ in response to HFD feeding and AOE supplementation. While PPAR γ mRNA levels mirrored CD36 expression, protein levels exhibited a reversed pattern. Previous research has also noted a reduction in hepatic PPAR γ protein content in HFD mice [49,50,51], with restoration upon intervention. Although PPAR γ primarily functions in adipose tissue to promote insulin sensitivity, its ambiguous function in the liver persists [52]. Post-transcriptional regulation may account for the discrepancy between mRNA and protein levels of PPAR γ in the liver. In summary, our results indicate that AOE reduces HFD-induced hepatic lipid deposition by downregulating genes implicated in lipid synthesis and uptake.

The regulation of hepatic iron content involves a dynamic interplay among processes of iron uptake, storage, and export [53]. In this investigation, supplementation with AOE successfully restored decreased hepatic iron deposition. When screening the genes affected by AOE, we discovered that under HFD conditions, TFR1, which mediates iron uptake, was significantly downregulated and FPN, which mediates iron efflux, was significantly upregulated. These changes were reversed by AOE treatment. The reduction in hepatic iron content induced by HFD was linked to a decrease in TFR1 expression and elevated FPN expression in the liver, indicating decreased iron absorption and increased iron release. Short-term high-fat feeding led to elevated hepatic iron levels, accompanied by decreased FPN expression [12] and increased TFR1 expression [12,16]. TFR1 expression levels showed a positive correlation with hepatic iron content, whereas FPN expression exhibited a negative correlation. Our findings are consistent with earlier studies showing that prolonged high-fat feeding increases FPN expression while reducing TFR1 expression in the liver. AOE reinstates hepatic iron levels through modulation of TFR1 and FPN, the key factors in iron uptake and efflux pathways.

Triterpenoids present in Alisma orientale have been demonstrated to modulate various nuclear transcription factors [32,54,55], including the glucocorticoid receptor (GR), FXR, retinoid receptor (RAR/RXR), and liver X receptor (LXR), all of which play crucial roles in influencing MAFLD [56]. Our findings support previous research indicating that a high-fat diet reduces FXR expression [57,58]. AOE completely restored FXR protein content in the liver. Consistent with prior studies, the administration of AOE counteracted the reduction in FXR protein levels caused by a Western diet [33]. FXR not only contributes to MAFLD development [25,26] but also regulates iron metabolism [27,28]. ChIP-PCR analysis revealed that FXR binding to the promoters of the SCD1 and FPN genes was significantly diminished in the high-fat (HF) group. AOE partly restored the decrease in FXR enrichment in the SCD1 and FPN gene promoters. Interestingly, we observed a negative correlation between FXR enrichment in promoters and gene expression. Consistent with previous findings suggesting that FXR binding sites may lose functionality in obesity, a substantial proportion of genes targeted by FXR are directly suppressed upon activation by its ligand [59]. These results suggest that AOE may activate FXR to control the expression of genes involved in lipid and iron metabolism in mice. This study has several limitations. Firstly, we employed a specific mouse model and a 60% high-fat diet, which raises uncertainty regarding the generalizability of our findings to other species or MAFLD models induced by different methods. Secondly, the study was cross-sectional in design, limiting our ability to establish a causal relationship between changes in iron and lipid metabolism. Thirdly, AOE is a complex mixture and, although previous studies have indicated that compounds such as alisol B [37] and alisol B 23-acetate [36,48] present in AOE may have preventive effects on MAFLD, it remains unclear as to which specific components are responsible and whether their effects are direct or indirect. Future research directions could focus on the following areas. Firstly, investigating the consistent significance of AOE on lipid and iron metabolism disorders during MAFLD progression across diverse animal models and induction methods. Secondly, assessing the therapeutic efficacy of individual components of AOE, such as alisol A, alisol B, alisol A 2,3-acetate, and alisol B 2,4-acetate, in treating high-fat diet-induced MAFLD models. Thirdly, utilizing hepatocyte lipid deposition models to determine if these components directly attenuate lipid accumulation. If successful, subsequent studies could aim to determine optimal dosages for potential drug screening in human MAFLD treatment.

## 5. Conclusions

In conclusion, AOE supplementation significantly reduced exorbitant lipid accumulation and recovered impaired iron balance in the liver of HFD-induced MAFLD mouse models. This effect may be mediated by the regulation of FXR enrichment in the promoters of the SCD1 and FPN genes. The reversal of liver iron imbalance by AOE represents a novel and previously undisclosed mechanism in the pathogenesis of MAFLD. These findings unveil a groundbreaking therapeutic role for AOE in the management of MAFLD.

## Figures and Tables

**Figure 1 nutrients-16-02272-f001:**
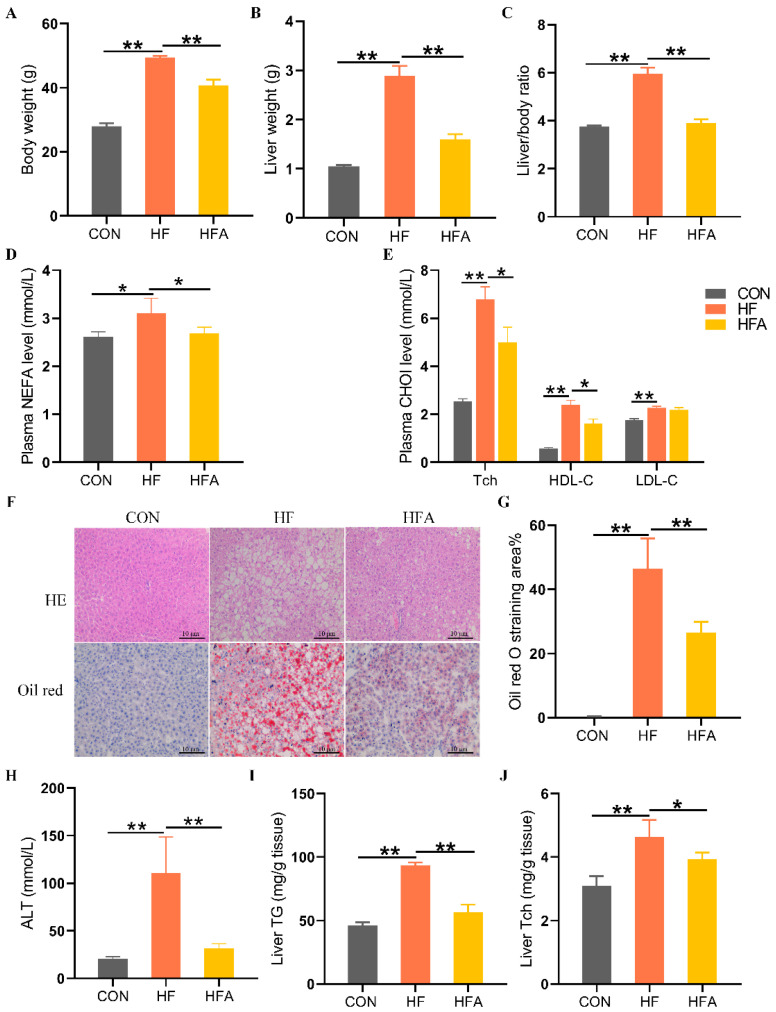
AOE mitigates hepatic lipid degeneration in mice. (**A**) Body weight. (**B**) Liver weight. (**C**) Liver index. (**D**) Plasma-free fatty acid levels. (**E**) Plasma cholesterol levels. (**F**) HE staining and Oil Red O staining. (**G**) Oil Red O staining score. (**H**) ALT. (**I**) Hepatic triglyceride content. (**J**) Hepatic cholesterol content. Median values are presented as medians ± interquartile ranges. Each group consists of *n* = 8 samples. * indicates *p* < 0.05 and ** indicates *p* < 0.01.

**Figure 2 nutrients-16-02272-f002:**
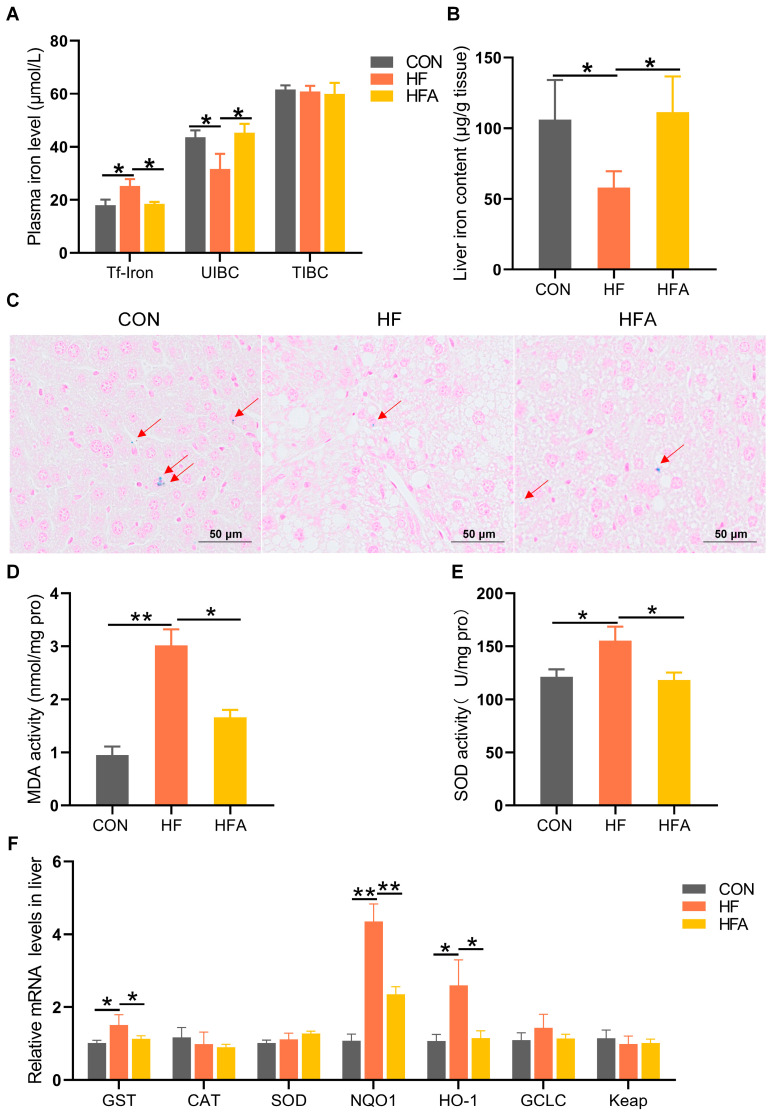
AOE alleviates hepatic iron metabolism dysregulation in mice. (**A**) Plasma iron content. (**B**) Iron content of the liver. (**C**) Prussian blue staining of the liver. The red arrows indicate the distribution of iron (**D**) Malondialdehyde content. (**E**) Superoxide dismutase (SOD) activity. (**F**) Oxidative stress-related genes. Median values are presented as medians ± interquartile ranges. Each group consists of *n* = 6 samples. * indicates *p* < 0.05 and ** indicates *p* < 0.01.

**Figure 3 nutrients-16-02272-f003:**
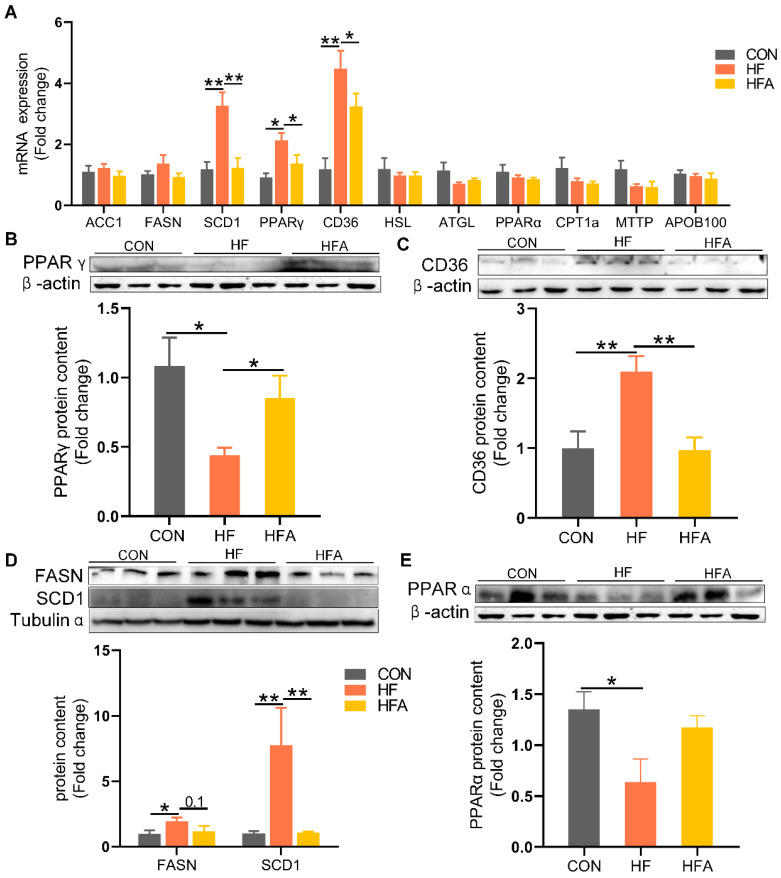
The regulatory effects of AOE on lipid metabolism genes. (**A**) Lipid metabolism gene levels. (**B**) PPAR γ protein content. (**C**) CD36 protein content. (**D**) De novo lipogenesis-related protein. (**E**) PPARα protein content. Median values are presented as medians ± interquartile ranges. Each group consists of *n* = 6 samples. * indicates *p* < 0.05 and ** indicates *p* < 0.01.

**Figure 4 nutrients-16-02272-f004:**
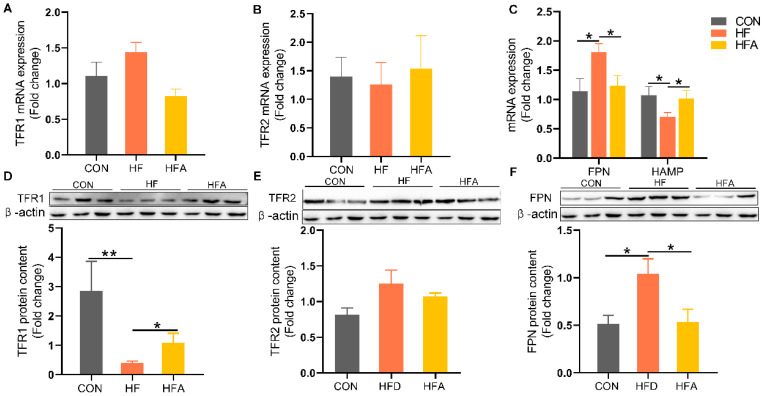
The regulatory effects of AOE on iron metabolism genes. (**A**) TFR1 mRNA levels. (**B**) TFR2 mRNA levels. (**C**) Iron efflux-related gene mRNA levels. (**D**) TFR1 protein content. (**E**) TFR2 protein content. (**F**) FPN protein content. Median values are presented as medians ± interquartile ranges. Each group consists of *n* = 6 samples. * indicates *p* < 0.05 and ** indicates *p* < 0.01.

**Figure 5 nutrients-16-02272-f005:**
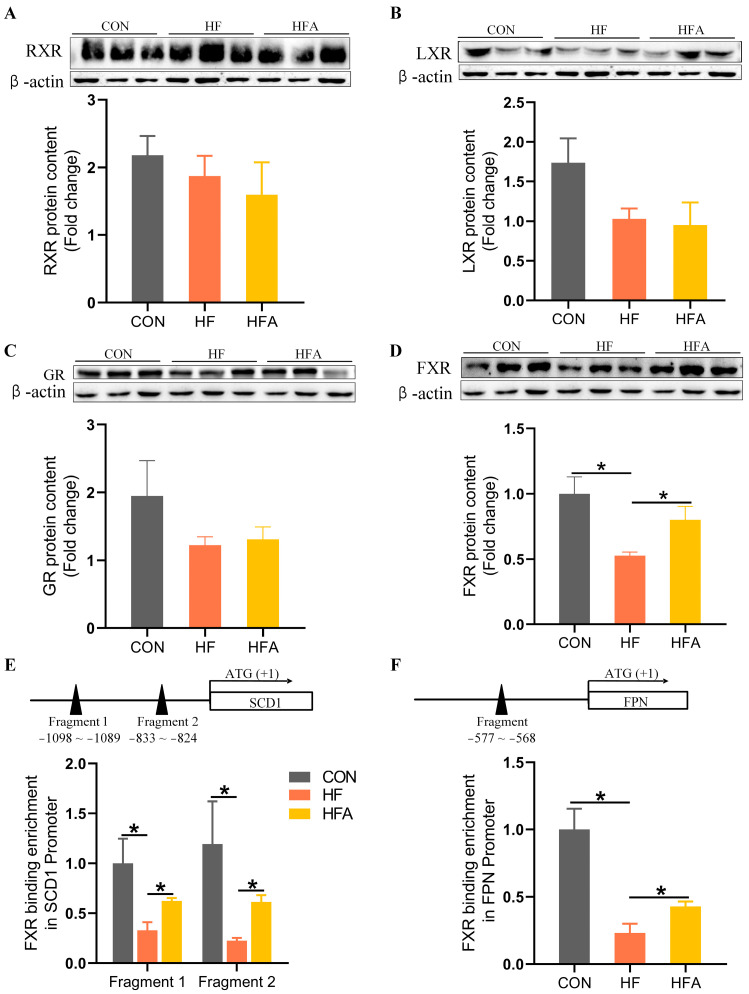
AOE promotes the enrichment of FXR regulatory factors in the promoter regions of iron metabolism and lipid metabolism genes. (**A**) RXR protein content. (**B**) LXR protein content. (**C**) GR protein content. (**D**) FXR protein content. (**E**) FXR enrichment in the SCD1 promoter region. (**F**) FXR enrichment in the FPN promoter region. Median values are presented as medians ± interquartile ranges. Each group consists of *n* = 6 samples. * indicates *p* < 0.05.

## Data Availability

The original contributions presented in the study are included in the article/Appendix A, further inquiries can be directed to the corresponding author.

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
