# Peer review of "Alisma Orientalis Extract Ameliorates Hepatic Iron Deregulation in MAFLD Mice via FXR-Mediated Gene Repression"

_nutrients, 2024, doi:10.3390/nu16142272_

Round 1

Reviewer 1 Report

Comments and Suggestions for Authors

  • Title: The title is clear and concise, indicating the focus on the impact of Alisma orientalis extract on lipid and iron metabolism in a MAFLD mouse model.
  • Abstract: The abstract should succinctly summarize the key findings and significance of the study. However, it is missing from the provided text. Ensure the abstract is included and highlights the study's objectives, methodology, key results, and conclusions.

Introduction

  • Background Information: The introduction provides a good background on the role of iron in the body and its importance in liver function and disease. It establishes the context well by discussing liver diseases and metabolic syndrome.
  • Gap in Knowledge: The introduction effectively identifies the gap in knowledge, specifically the unclear role of FXR in liver lipid and iron metabolism and the potential of Alisma orientalis in MAFLD treatment.
  • Hypothesis and Objectives: The hypothesis that Alisma orientalis extract can alleviate MAFLD by regulating lipid and iron metabolism through FXR activation is clear. The objectives are stated at the end of the introduction, providing a clear direction for the study.

Materials and Methods

  • Detail and Reproducibility: The methods section is detailed and provides sufficient information for reproducibility. The procedures for plant extraction, animal treatment, biochemical assays, histological evaluations, and molecular analyses are well described.
  • Experimental Design: The experimental design, including the control and high-fat diet groups, is appropriate for studying the effects of Alisma orientalis extract. The use of various assays and techniques strengthens the study.
  • Statistical Analysis: The statistical analysis methods are briefly mentioned. However, more details on the specific tests used and their appropriateness for the data type should be provided.

Results

  • Presentation of Data: The results are clearly presented with figures and tables to support the findings. However, ensure all figures and tables are properly labeled and referenced in the text.
  • Data Interpretation: The interpretation of the results should be more comprehensive. For example, explain why AOE might reduce lipid droplet accumulation or alter iron metabolism in the context of FXR activation.
  • Statistical Significance: Statistical significance is reported, but ensure that all reported p-values are consistent and properly annotated in the figures and tables.

Discussion

  • Interpretation of Findings: The discussion should provide a deeper interpretation of the findings, linking them back to the objectives and hypotheses. Compare the results with existing literature to highlight the novelty and significance of the study.
  • Limitations: The limitations of the study are not discussed. Address potential limitations, such as the use of only one animal model or the specific conditions of the high-fat diet.
  • Future Directions: Suggest future research directions based on the findings. For instance, exploring the molecular mechanisms of FXR activation by Alisma orientalis extract in more detail.

Conclusion

  • Summary of Findings: The conclusion should summarize the key findings and their implications for MAFLD treatment. Emphasize the potential of Alisma orientalis extract as a therapeutic agent.
  • Practical Implications: Discuss the practical implications of the findings, such as potential clinical applications and the next steps for drug development.

References

  • Completeness: Ensure all references are complete and follow a consistent citation style. Verify that all cited works are relevant and up-to-date.

Decision

Based on the critical review, the paper is well-structured and provides valuable insights into the role of Alisma orientalis extract in MAFLD treatment. However, improvements in data interpretation, discussion of limitations, and a more detailed statistical analysis are needed. I recommend minor revisions before the paper can be accepted for publication.

Areas for Improvement

  1. Abstract: Include a concise abstract summarizing the study.
  2. Statistical Analysis: Provide more details on the statistical tests used.
  3. Data Interpretation: Enhance the interpretation of results and link them to the hypothesis.
  4. Discussion: Discuss the limitations and compare findings with existing literature.
  5. Future Directions: Suggest future research directions and practical implications.
  6. References: Ensure all references are complete and properly formatted.

Overall, with these revisions, the paper will make a significant contribution to the field of metabolic liver diseases and the therapeutic potential of traditional Chinese medicine.

Comments on the Quality of English Language

Minor changes in English language must be done. 

Author Response

Reviewer:

Areas for Improvement

  1. Abstract: Include a concise abstract summarizing the study.

Reply: Abstract: Iron is a vital trace element for our bodies, and its imbalance can lead to various diseases. The progression of metabolic-associated fatty liver disease (MAFLD) is often accompanied by disturbances in iron metabolism. Alisma orientale extract (AOE) has been reported to alleviate MAFLD. However, research on its specific lipid metabolism targets and its potential impact on iron metabolism during the progression of MAFLD remains limited. To establish a model of MAFLD, mice were fed either a standard diet (CON) or a high-fat diet (HFD) for 9 weeks. The mice raised on the HF diet were then randomly assigned to the HF group and the HFA group, with the HFA group receiving AOE by gavage on a daily basis for 13 weeks. Supplementation with AOE significantly reduced excessive lipid deposition in the liver and restored the iron content of the liver. AOE partially but significantly reversed dysregulated lipid metabolizing genes (SCD1, PPAR γ, CD36) and iron metabolism genes (TFR1, FPN, HAMP) induced by HFD. Chromatin immunoprecipitation assays indicated that the reduced enrichment of FXR on the promoters of SCD1 and FPN genes induced by HFD was significantly reversed by AOE. These findings suggest that AOE may alleviate HFD-induced disturbances in liver lipid and iron metabolism through FXR-mediated gene repression.

  1. Statistical Analysis: Provide more details on the statistical tests used.

Reply: Using Kolmogorov-Smirnov test in SPSS 20.0 software (SPSS Inc., Chicago, IL, USA) to assess normal distribution of our data or log-transformed data, we confirmed that the results met the criteria for normal distribution. Subsequently, one-way analysis of variance (ANOVA) with Least Significant Difference (LSD) test was conducted to determine significant differences between group means. Data are presented as mean ± standard error of the mean (SEM). Statistical significance is defined at #, *P < 0.05 and ##, **P < 0.01 levels.

  1. Data Interpretation: Enhance the interpretation of results and link them to the hypothesis.

Reply: Thank you very much for your suggestion. We have further interpreted our results in each section, establishing closer connections with our hypotheses.

  1. Discussion: Discuss the limitations and compare findings with existing literature.

Reply: This study has several limitations. First, we used a specific mouse model and a 60% high-fat diet, so it is unclear whether our results can be replicated in other species or in MAFLD models established by different methods. Second, the study was designed as a cross-sectional study; while we observed changes in iron metabolism and lipid metabolism, we could not determine the causal relationship between them. Third, AOE is a mixture, and although previous studies have reported that alisol B[37] and alisol B 23-acetate[36,48] contained in AOE can have preventive effects on MAFLD, it remains unclear which specific components of AOE are functional and whether these effects are direct or indirect.

  1. Future Directions: Suggest future research directions and practical implications.

Reply: In future research, we can focus on the following directions: First, by using various animal models and different MAFLD induction methods, we can investigate whether the effects of AOE on lipid metabolism disorder and iron metabolism disorder are universally significant in the progression of MAFLD. Second, we can treat high-fat diet-induced MAFLD models with different monomers contained in AOE, such as alisol A, alisol B, alisol A 2,3-acetate, and alisol B 2,4-acetate, to determine which specific substances are effective. Third, we can use hepatocyte lipid deposition models to study whether these substances directly alleviate lipid accumulation. If they do, we can explore the appropriate dosages, thereby providing effective alternatives for drug screening in human MAFLD.

  1. References: Ensure all references are complete and properly formatted.

Reply: Thank you very much for your suggestion. We have rechecked the references to ensure the completeness and correct formatting of the citation materials.

Reviewer 2 Report

Comments and Suggestions for Authors

Yanlin Li and colleagues have conducted an experimental study in mice to evaluate the effect of Alisma orientale extract (AOE) on iron homeostasis in the liver of mice with metabolic-associated fatty liver disease (MAFLD) induced by high-fat diets. Mice were fed either a standard diet (CON; n = 8) or a high-fat diet (HFD; n = 16) for 9 weeks. Mice raised on the HF diet (we assume the authors are referring to the HFD) were then randomly assigned to the HF group and the HFA group, with the HFA group receiving AOE by gavage daily for 13 weeks. The authors concluded that AOE supplementation significantly reduced excessive lipid accumulation and restored iron homeostasis in the liver of HFD-induced MAFLD mouse models. This effect may be mediated by upregulation of farnesoid X receptor (FXR) enrichment at the promoters of SCD1 and FPN genes.

While these findings are interesting, several methodological and analytical issues are of concern:

Major comments

The statistical analysis is erratic and confusing. The only statistical analysis they describe is the comparison of two means by t-test. First, the sample sizes are very small (eight mice in each of the three groups). Therefore, the summary in means is not adequate as the data could have skewness. Eight data per group do not allow testing for normality (without skewness). The arithmetic mean is very sensitive to outliers, which is not the case with the median. With eight observations, a single outlier would produce changes in the mean that may have no biological interpretation. For these same reasons, the t-test may be spurious. Therefore, the data would be summarized as medians and interquartile ranges (25th -75th percentiles). Furthermore, to compare three groups, a single test should be used first. If means were to be compared, it would be the F-test of the analysis of variance. Since it is more appropriate to sum the variables into medians, the Kruskal-Wallis test would be the most appropriate. If the Kruskal-Wallis test showed statistical significance, the comparison of pairs of groups would have to be done by multiple comparisons.

Minor comments

1.     1.         The presentation of results using bar charts is cumbersome and takes up a lot of space in the manuscript. I suggest that the authors present the results in tables with five columns. The first column would correspond to the variable identifiers. The next three columns would show the medians and interquartile ranges of the corresponding variable in each study group and, finally, the last column would show the p-value corresponding to the Kruskal-Wallis test. Under the significance assumption of the Kruskal-Wallis test, the resulting multiple comparisons can be summarized by superscripts. Authors could search Google Scholar for the following phrase: "different superscripts indicate significant differences for p < 0.05".

2.     However, the authors could present the results using bar charts, but not for means and standard errors, but for medians and interquartile ranges. This can be done with the R package, ggplot2 library.

3.     In the text there is confusion with the group of 16 mice. Both in the abstract and in line 125 it is referred to as HFD (High-far diet). However, in line 20 it is indicated: “The mice raised on the HF diet were then randomly 20 assigned to the HF group and the HFA group”. Obviously it should say: “The mice raised on the HFD diet …”. Please make corrections in all appropriate places.

4.     Line 36: change “Liver” to “liver”. 

5.     Line 81: what does (Sam.) mean?

6.     When the authors refer to the “Alisma orientalis extract” throughout the manuscript, they sometimes refer to it as Alisma, sometimes as “AOE” and sometimes as “Alisma orientalis extract (AOE)”. Please unify the expressions.

7.     Table 1 should be divided into two. The first one would contain the percentages of grams and kilocalories and the other the quantities.

8.     In Table 1, what are the components of gm missing to reach 100%?

9.     The results described in section 3.1 should be modified as indicated in the “major comments” section. The authors make comparisons between pairs of treatments without taking into account multiple comparisons, which is erratic as indicated above.

10.  Figure 1.F is misplaced.

11.  The legend in Figures 1 through 4 is confusing. I think the symbols "#" and "##" refer to comparisons of the “HF” group with the “CON”, while "*" and "**" refer to comparisons of the “AOE” group with the “HF”. Please clarify.

Author Response

Reviewer:

MAJOR COMMENTS

The statistical analysis is erratic and confusing. The only statistical analysis they describe is the comparison of two means by t-test. First, the sample sizes are very small (eight mice in each of the three groups). Therefore, the summary in means is not adequate as the data could have skewness. Eight data per group do not allow testing for normality (without skewness). The arithmetic mean is very sensitive to outliers, which is not the case with the median. With eight observations, a single outlier would produce changes in the mean that may have no biological interpretation. For these same reasons, the t-test may be spurious. Therefore, the data would be summarized as medians and interquartile ranges (25th -75th percentiles). Furthermore, to compare three groups, a single test should be used first. If means were to be compared, it would be the F-test of the analysis of variance. Since it is more appropriate to sum the variables into medians, the Kruskal-Wallis test would be the most appropriate. If the Kruskal-Wallis test showed statistical significance, the comparison of pairs of groups would have to be done by multiple comparisons.

Reply: Thank you for your question regarding the statistical methods used in our study. We understand that using the t-test for group comparisons has its limitations, especially with our small sample size. Here is an explanation of our methodology. In selecting the sample size, we conducted a thorough statistical power analysis using G*Power software and confirmed that this sample size (N=8) achieves sufficient statistical power to ensure the reliability of our experimental results. The prerequisite for t-tests or one-way ANOVA is that the data meet the normal distribution assumption. Therefore, we first performed the Kolmogorov-Smirnov test on our data or log-transformed data to analyze normal distribution. Only after confirming the normality assumption did we proceed with further statistical analyses. According to our analysis, we were able to draw conclusions at the required significance level, demonstrating the reliability and reproducibility of our experimental outcomes.

MINOR COMMENTS

  1. The presentation of results using bar charts is cumbersome and takes up a lot of space in the manuscript. I suggest that the authors present the results in tables with five columns. The first column would correspond to the variable identifiers. The next three columns would show the medians and interquartile ranges of the corresponding variable in each study group and, finally, the last column would show the p-value corresponding to the Kruskal-Wallis test. Under the significance assumption of the Kruskal-Wallis test, the resulting multiple comparisons can be summarized by superscripts. Authors could search Google Scholar for the following phrase: "different superscripts indicate significant differences for p < 0.05".

Reply: Thank you very much for your advice. However, to make the presentation more aesthetically pleasing, and easier for readers to understand, we prefer to use a bar chart to display our results.

  1. However, the authors could present the results using bar charts, but not for means and standard errors, but for medians and interquartile ranges. This can be done with the R package, ggplot2 library.

Reply: Thank you very much for your suggestion. In our current research, we often habitually use mean values and standard errors to present our results. This method of presentation may not be as rigorous as the one you suggested, but it is feasible for our current publications. Several similar studies that have already been published use this method. Here are some examples of the articles:

  1. Li Y, Jiang W, Feng Y, Wu L, Jia Y, Zhao R. Betaine Alleviates High-Fat Diet-Induced Disruption of Hepatic Lipid and Iron Homeostasis in Mice. Int J Mol Sci. 2022 Jun 3;23(11):6263. doi: 10.3390/ijms23116263. PMID: 35682942; PMCID: PMC9180950.
  2. Sun B, Jia Y, Yang S, Zhao N, Hu Y, Hong J, Gao S, Zhao R. Sodium butyrate protects against high-fat diet-induced oxidative stress in rat liver by promoting expression of nuclear factor E2-related factor 2. Br J Nutr. 2019 Aug 28;122(4):400-410. doi: 10.1017/S0007114519001399. PMID: 31204637.
  3. Sun B, Jia Y, Hong J, Sun Q, Gao S, Hu Y, Zhao N, Zhao R. Sodium Butyrate Ameliorates High-Fat-Diet-Induced Non-alcoholic Fatty Liver Disease through Peroxisome Proliferator-Activated Receptor α-Mediated Activation of β Oxidation and Suppression of Inflammation. J Agric Food Chem. 2018 Jul 25;66(29):7633-7642. doi: 10.1021/acs.jafc.8b01189. Epub 2018 Jul 11. PMID: 29961332.
  4. Songtrai S, Pratchayasakul W, Arunsak B, Chunchai T, Kongkaew A, Chattipakorn N, Chattipakorn SC, Kaewsuwan S. Cyclosorus terminans Extract Ameliorates Insulin Resistance and Non-Alcoholic Fatty Liver Disease (NAFLD) in High-Fat Diet (HFD)-Induced Obese Rats. Nutrients. 2022 Nov 19;14(22):4895. doi: 10.3390/nu14224895. PMID: 36432581; PMCID: PMC9693870.

  1. In the text there is confusion with the group of 16 mice. Both in the abstract and in line 125 it is referred to as HFD (High-far diet). However, in line 20 it is indicated: “The mice raised on the HF diet were then randomly 20 assigned to the HF group and the HFA group”. Obviously it should say: “The mice raised on the HFD diet …”. Please make corrections in all appropriate places.

Reply: Thank you very much for your suggestion. I have made the necessary corrections.

  1. Line 36: change “Liver” to “liver”.

Reply: Thank you very much for your suggestion. I have made the necessary corrections.

  1. Line 81: what does (Sam.) mean?

Reply: SAM usually stands for Standardized Alisma Extract, referring to a standardized extract of Alisma. This abbreviation is commonly used in the fields of medicine and dietary supplements. Alisma orientalis is the botanical name for the plant known as 泽泻 (zé xiè) in Chinese. In summary, Alisma orientalis is the official botanical name, while it is called SAM because it is the abbreviation for the standardized extract of Alisma. These different names are used in various contexts and fields to facilitate precise identification and communication.

  1. When the authors refer to the “Alisma orientalis extract” throughout the manuscript, they sometimes refer to it as Alisma, sometimes as “AOE” and sometimes as “Alisma orientalis extract (AOE)”. Please unify the expressions.

Reply: Thank you very much for your suggestion. I have made the necessary corrections.

  1. Table 1 should be divided into two. The first one would contain the percentages of grams and kilocalories and the other the quantities.

Reply: Thank you very much for your suggestion. I have made the necessary corrections.

CON

HF

Energizing material

gm%

Kcal%

gm%

Kcal%

Protein

19.2

20

26.2

20

Carbohydrate

67.3

70

26.3

20

Fat

4.3

10

34.9

60

Total

100

100

3.85

5.24

Ingredient

gm

Kcal

gm

Kcal

Casein, 80 Mesh

200

800

200

800

L-cystine

3

12

3

12

Corn starch

315

1260

0

0

Maltodextrin 10

35

140

125

500

Sucrose

350

1400

68.8

275.2

Cellulose

50

0

50

0

Soybean Oil

25

255

25

255

Lard

20

180

245

2205

Mineral mix S10026

10

0

10

0

Dicalcium Phosphate

13

0

13

0

Calcium Carbonate

5.5

0

5.5

0

Potassium Citrate,1 H20

16.5

0

16.5

0

Vitamin Mix V10001

10

40

10

40

Choline Bitartrate

2

0

2

0

Pigment

0.05

0

0.05

0

Total

1055.05

4057

773.85

4057

  1. In Table 1, what are the components of gm missing to reach 100%?

Reply: The composition of food includes proteins, fats, and carbohydrates, but they are not the only components. Food also contains water, dietary fiber, minerals, microorganisms, enzymes, organic acids, pigments, and flavoring agents. Although these components do not provide energy, they are essential for the normal functioning of the body. This is why the total percentage of proteins, fats, and carbohydrates does not add up to 100%.

  1. The results described in section 3.1 should be modified as indicated in the “major comments” section. The authors make comparisons between pairs of treatments without taking into account multiple comparisons, which is erratic as indicated above.

Reply: Thank you very much for your suggestion. As you mentioned, in our study, we first used a multifactorial approach to account for multiple comparisons among the three groups, and subsequently conducted pairwise comparisons (One-way ANOVA and T-tests) upon identifying differences. Indeed, this is exactly what we did. We initially conducted multiple comparisons on the data and then performed pairwise comparisons for groups that showed significant differences. Since our primary focus was on comparing the (CON vs. HF) and (HF vs. HFA) groups, we only mentioned these specific comparisons in the results description. We believe that misunderstanding may have arisen due to our overly simplified description of the statistical methods. Therefore, we will supplement our description of the statistical methods to make it more detailed and clear.

  1. Figure 1.F is misplaced.

Reply: Thank you very much for your suggestion. I have made the necessary corrections.

  1. The legend in Figures 1 through 4 is confusing. I think the symbols "#" and "##" refer to comparisons of the “HF” group with the “CON”, while "*" and "**" refer to comparisons of the “AOE” group with the “HF”. Please clarify.

Reply: That's correct. We use “#” and “##” to refer to comparisons between the HF group and the CON group, and “*” and “**” to refer to comparisons between the HFA (AOE) group and the HF group. This method of notation has been reported in previously published articles

  1. Li Y, Jiang W, Feng Y, Wu L, Jia Y, Zhao R. Betaine Alleviates High-Fat Diet-Induced Disruption of Hepatic Lipid and Iron Homeostasis in Mice. Int J Mol Sci. 2022 Jun 3;23(11):6263. doi: 10.3390/ijms23116263. PMID: 35682942; PMCID: PMC9180950.
  2. Nagappan A, Jung DY, Kim JH, Jung MH. Protective Effects of Gomisin N against Hepatic Cannabinoid Type 1 Receptor-Induced Insulin Resistance and Gluconeogenesis. Int J Mol Sci. 2018 Mar 23;19(4):968. doi: 10.3390/ijms19040968. PMID: 29570673; PMCID: PMC5979504.

Round 2

Reviewer 1 Report

Comments and Suggestions for Authors

The article shows improvement with the changes made. I would accept it for publication in its current version.

Comments on the Quality of English Language

Minor English language must be done. 

Author Response

Reviewer:

Minor English language must be done

Reply: Thank you very much for your suggestions. We have made the necessary minor English language corrections in the manuscript, with all changes highlighted in yellow.

Reviewer 2 Report

Comments and Suggestions for Authors

In the previous review I indicated to the authors that the captions in Figures 1 through 4 were confusing, but the authors have done nothing to clarify this. It is possible that the authors' intent is to compare the “HFA” group to the “CON” and the “HFA” to the “HF”. In such a case, the symbols "#" and "##" should be above the bar corresponding to the “CON” group and the symbols "*" and "**" above the one corresponding to the “HF” group.

Author Response

Reviewer:

In the previous review I indicated to the authors that the captions in Figures 1 through 4 were confusing, but the authors have done nothing to clarify this. It is possible that the authors' intent is to compare the “HFA” group to the “CON” and the “HFA” to the “HF”. In such a case, the symbols "#" and "##" should be above the bar corresponding to the “CON” group and the symbols "*" and "**" above the one corresponding to the “HF” group.

Reply: I sincerely apologize for misunderstanding your point. We have made changes to the captions in the figures.

Figure 1:

Figure 1. AOE mitigates hepatic lipid degeneration in mice. (A) Body weight. (B) Liver weight. (C) Liver index. (D) Plasma free fatty acid levels. (E) Plasma cholesterol levels. (F) HE staining and Oil Red O staining. (G) Oil Red O staining score. (H) Hepatic triglyceride content. (I) Hepatic cholesterol content. Median values are presented as medians ± interquartile ranges. Each group consists of n = 8 samples. * indicates p < 0.05 and ** indicates p < 0.01.

Figure 2:

Figure 2. AOE alleviates hepatic iron metabolism dysregulation in mice. (A) Plasma iron content. (B) Iron content of the liver. (C) Prussian blue staining of the liver. (D) Malondialdehyde content. (E) Superoxide dismutase(SOD)activity. (F) Oxidative stress-related genes. Median values are presented as medians ± interquartile ranges. Each group consists of n = 6 samples. * indicates p < 0.05 and ** indicates p < 0.01.

Figure 3:

Figure 3. The regulatory effects of AOE on lipid metabolism genes. (A) Lipid metabolism gene levels. (B) PPAR γ protein content. (C) CD36 protein content. (D) De novo lipogenesis-related protein. (E) PPARα protein content. Median values are presented as medians ± interquartile ranges. Each group consists of n = 6 samples. * indicates p < 0.05 and ** indicates p < 0.01.

Figure 4:

Figure 4. The regulatory effects of AOE on iron metabolism genes. (A) TFR1 mRNA levels. (B) TFR2 mRNA levels. (C) Iron efflux-related gene mRNA levels. (D) TFR1 protein content. (E) TFR2 protein content. (F) FPN protein content. Median values are presented as medians ± interquartile ranges. Each group consists of n = 6 samples. * indicates p < 0.05 and ** indicates p < 0.01.

Figure 5:

Figure 5. AOE promotes enrichment of FXR regulatory factors in the promoter regions of iron metabolism and lipid metabolism genes. (A) RXR protein content. (B) LXR protein content. (C) GR protein content. (D) FXR protein content. (E) FXR enrichment in the SCD1 promoter region. (F) FXR enrichment in the FPN promoter region. Median values are presented as medians ± interquartile ranges. Each group consists of n = 6 samples. * indicates p < 0.05 and ** indicates p < 0.01.
